# Application of Navigated Transcranial Magnetic Stimulation (nTMS) to Study the Visual–Spatial Network and Prevent Neglect in Brain Tumour Surgery

**DOI:** 10.3390/cancers16244250

**Published:** 2024-12-20

**Authors:** Camilla Bonaudo, Elisa Castaldi, Agnese Pedone, Federico Capelli, Shani Enderage Don, Edoardo Pieropan, Andrea Bianchi, Marika Gobbo, Giuseppe Maduli, Francesca Fedi, Fabrizio Baldanzi, Simone Troiano, Antonio Maiorelli, Giovanni Muscas, Francesca Battista, Luca Campagnaro, Serena De Pellegrin, Andrea Amadori, Enrico Fainardi, Riccardo Carrai, Antonello Grippo, Alessandro Della Puppa

**Affiliations:** 1Neurosurgery, Department of Neuroscience, Psychology, Pharmacology and Child Health, University Hospital of Careggi, University of Florence, 50134 Florence, Italy; agnese.pedone@unifi.it (A.P.); federico.capelli@unifi.it (F.C.); shani.enderagedon@unifi.it (S.E.D.); fedifr@aou-careggi.toscana.it (F.F.); muscasgi@aou-careggi.toscana.it (G.M.); francesca.battista@unifi.it (F.B.); campagnarol@aou-careggi.toscana.it (L.C.); alessandro.dellapuppa@unifi.it (A.D.P.); 2Department of Neuroscience, Psychology, Pharmacology, and Child Health, University of Florence, 50121 Florence, Italy; elisa.castaldi@unifi.it (E.C.); angus.gius@gmail.com (G.M.); 3Department of Information Engineering, University of Padua, 35122 Padua, Italy; edoardo.pieropan@gmail.com; 4Antares S.p.A., Dueville, 36031 Vicenza, Italy; 5Neuroradiology, Department of Biomedical, Experimental and Clinical Sciences, University of Florence, 50121 Florence, Italy; dottabianchi@gmail.com (A.B.); enrico.fainardi@unifi.it (E.F.); 6Department of Life Sciences, University of Trieste, 34127 Trieste, Italy; gobbomarika@gmail.com; 7Cognitive Neuroscience Laboratory, University of Udine, 33100 Udine, Italy; 8Neurophysiopathology Unit, University Hospital of Careggi, University of Florence, 50134 Firenze, Italy; baldanzifabrizio@gmail.com (F.B.); troianos@aou-careggi.toscana.it (S.T.); maiorellia@aou-careggi.toscana.it (A.M.); riccardo.carrai@unifi.it (R.C.); antonello.grippo@unifi.it (A.G.); 9DIDAS Medicine, Neurology Clinic, University Hospital of Padua, 35128 Padua, Italy; serena.depellegrin@unipd.it; 10Neuroanesthesia and Intensive Care, University Hospital of Careggi, 50134 Florence, Italy; andreaamadori4@virgilio.it

**Keywords:** neuro-oncology, awake surgery, nTMS, neglect, visuospatial network, tractography, cognitive mapping, brain surgery

## Abstract

Visuospatial circuits (VS) must be preserved during brain tumours’ excision. Navigated TMS is a valid tool to preoperatively map VS networks and to optimise functional preservation. In this work, we tested 27 patients with nTMS preoperative mapping and an experimental test for VS abilities, called VISA, demonstrating a good clinical outcome (functional recovery 80–98.86%) and a useful nTMS map for tractography reconstruction (with the main involvement of the second and the third branches of the superior longitudinal fasciculus). Finally, a comparison of nTMS and DCS points in awake surgery (n = 10 patients) documented a sensitivity (Se) of 12%, a specificity (Sp) of 91.21%, a positive predictive value (PPV) of 42%, a negative predictive value (NPV) of 66%, and an accuracy of ~63.7%. According to our preliminary results, nTMS is advantageous to study cognitive functions, minimising neurological impairment. Further analyses are needed to validate our data.

## 1. Introduction

Modern oncological neurosurgery is aimed at sparing pivotal functions, providing a complete neurocognitive assessment [1]. Visuospatial function (VSf) is essential for everyday life. Unilateral spatial neglect (USN) is an acquired disturbance characterised by difficulty in directing attention to meaningful stimuli presented in the space contralateral to the brain lesion, not linked to sensory or motor deficits [2], and can be assessed by administering specific tests developed in neuropsychology. Navigated TMS (nTMS) has been recently proposed in neuro-oncological research to detect brain sites functionally involved in cognitive processes [3,4] to preoperatively map cortical areas involved in language [5], arithmetic calculation [6], and, more recently, VSf [7].

Nowadays, together with direct cortical stimulation (DCS), the gold standard to map cortical functions in awake surgery, repetitive nTMS (rnTMS) is routinely used for preoperative mapping, guiding tumour resection either in asleep or awake surgery [8,9,10,11]. The joint use of nTMS and DCS can provide useful information on the convergence between these two techniques and the reliability of nTMS for mapping VS attention before and even after surgery. Our research group has recently dedicated attention to arithmetic calculation nTMS mapping to improve cognitive studies and the clinical outcomes of oncological patients [6], and this new work aimed at adding a new component to preoperative planning. As far as we know, in neuro-oncological neurosurgical patients, USN is usually monitored to avoid postoperative impairments, allow a good quality of life after surgery, and possibly return to a normal life. Neuropsychological evaluation, nTMS cortical data, and DTI tractography can be integrated in preoperative planning to study multiple functions, guiding surgery, customising the surgical approach, and orienting the postoperative rehabilitation process.

On the basis of these considerations, we defined the main objectives of this work as follows. The primary aim consisted of adding VS perception, reasoning, and attention testing within preoperative mapping with nrTMS to evaluate the pre-/postoperative clinical outcome of patients operated on for brain lesions, including awake and asleep procedures. The secondary aim consisted of evaluating the possible subcortical white matter fibre tracts involved in the VS network (comparing tractographies derived from anatomical tensors and from nTMS data). The third aim consisted of evaluating the correspondence between nTMS and DCS points during awake surgery to determine nTMS’s reliability for study and preservation of the VS network.

## 2. Materials and Methods

We designed a monocentric, prospective clinical study, collecting data from patients operated on at the University Hospital of Careggi, Florence, Italy, who presented focal lesions in eloquent areas for language, arithmetic calculation, and/or visuospatial functions. Preoperative planning included a neuropsychological/logopaedic evaluation, MRI imaging, and an nTMS protocol including cognitive functions, such as language (L), calculation (C), and neglect (N). All operations were performed by the same senior neurosurgeon (A.D.P).

The cognitive evaluation included a MiniMental State Examination (MMSE), the Aachener Aphasie Test (AAT), the neuropsychological test for aphasia (ENPA), arithmetic items for calculations, VS abilities, the presence of USN through barrage tests (i.e., the broken hearts test) taken from the translated Italian version of the Oxford Cognitive Screen (OCS), the bells test, and the clock drawing test (Shulman correction).

### 2.1. Navigated TMS Procedure for VSf

Visuospatial integrated assessment (VISA) involved an experimental test evaluating multiple cognitive functions simultaneously. Dr Castaldi and her colleagues are currently developing a test, inspired by Dehaene et al. [12], to simultaneously evaluate visuospatial perception, visuospatial attention, non-verbal reasoning, cognitive control, and number reading. This test was proposed to our patients after classical neuropsychological tests for neglect (classic tests: line bisection test, OCS, and bell test) (Figure 1).

### 2.2. Preoperative MRI

Preoperative magnetic resonance imaging (MRI) was performed at our neuroradiological department, using a 3T Tesla MRI machine (Ingenia 3T, Philips Medical Systems, Best, The Netherlands). Diffusion tensor imaging (DTI) sequences were employed for tractographies. Multiple regions of interest (ROIs) were identified and used to create subcortical pathways, using IntelliSpace Portal software 9.0 (Philips, Best, The Netherlands). To reproduce the three branches of the superior longitudinal fasciculus (SLFI, -II, and -III), the method previously described [13] was applied to identify these. Our expert neuroradiologist (A.B.) checked out all the tractographies and verified the corresponding projection of the cortical nTMS spots to identify the branches of the SLF principally involved in VS function. The same SLF branches were reconstructed from the nTMS stimulation spots and verified in comparison with the ones anatomically drawn.

### 2.3. Preoperative nrTMS Mapping

Preoperative cognitive mapping was performed, including L, C and VSf with the rnTMS system (Galileo NetBrain Neuronavigator 9000, EB Neuro Corp., Florence, Italy). The mapping method has previously been described [6]. The experimental integrated VS test, which is potentially able to detect USN, was adapted for tablet use to facilitate practicality in an nTMS-dedicated room or in the operating setting. Participants were asked to identify the odd one out among four drawings, while disregarding the surrounding square or circular frame, and responding by spelling out the number corresponding to the selected drawing (Figure 1). All tests were displayed on a screen (picture presentation time, 4 s; inter-picture interval, 1 s) in front of the patient, who answered aloud with 5 pulses of 5 Hz nrTMS perturbation. VSfs were mapped on 70 spots over the parietal, temporal, and frontal cortices. A cortical spot was considered positive for any function when nrTMS perturbation induced an error in at least two out of three trials [14]. An MRI DTI scan was used to create an nTMS-based-tractography [15,16], to be compared with the “anatomical” tractography, based on tensor reconstruction, showing the SLF’s branches.

Intraoperative monitoring was performed either during awake or asleep procedures; motor and somato-sensorial evoked potentials (MEPs/SSEPs), EEG, and electrocorticography were recorded. In awake surgeries, the Penfield paradigm for the applied intensity of DCS was between 0.5 and 3 mA.

### 2.4. Anaesthesiologic Parameters for Awake Surgery

Sedation was performed with 2% propofol in the TCI Schnider model, with a target effect site concentration (CET) of 1–1.5 µg/mL, and remifentanil (0.02–0.05 µg/kg/min) in spontaneous breathing. ECG monitoring, invasive blood pressure, oxygen saturation %, respiratory rate, EtC02, and diuresis output were controlled. For the scalp nerve block, a Mayfield application of 2% lidocaine, 0.5% bupivacaine (5 mg/mL) with epinephrine, and 7.5 mg/mL ropivacaine were administered. During the awake phase and cortical mapping, BIS was maintained over a value of 90 [17]; a continuous EEG pattern was recorded to prevent seizures.

### 2.5. Identification of nTMS Points for VSf and Neglect

We identified points for VSf and USN in the parietal lobe of either the left or right lesions, mainly located on the supramarginal gyrus (SMG), angular gyrus (AG), or the superior temporal gyrus (STG). Using the neuronavigation system integrated in the microscope (Kinevo Zeiss 900, Carl Zeiss, Oberkochen, Germany), we identified nTMS spots for VS functions and marked them with sterile tags on the brain’s cortical surface to guide surgical resection, according to functional boundaries in asleep procedures. During awake surgeries, the Penfield bipolar stimulation technique was applied (current intensity between 0.5 and 3 mA) for VSf mapping.

The surgical strategy was modulated on the basis of nTMS preoperative mapping, sparing positive points on the cortical surface whenever possible and considering the areas with negative nTMS points to be useful for safe access to brain lesions.

### 2.6. Statistical Analyses of nTMS/DCS Positive Points 

We used the method previously proposed in our recent work on nTMS for monitoring arithmetic calculation [6]. We applied a statistical evaluation considering the 3D brain location involved with VSf; we based all the analyses on the DCS points collected during awake procedures and the nTMS spatial points’ coordinates (x,y,z) provided by the neuronavigational system (StealthStation Medtronic, Minneapolis, MN, USA) [6]. All analyses were performed using specific scripts written in Python and the mathematical statistical method previously elaborated [6].

### 2.7. Ethics

The study was approved by the Ethics Committee “Comitato Etico Area Vasta Centro” (CEAVC Protocol No. 17003), in compliance with the Declaration of Helsinki.

## 3. Results

Between April 2023 and March 2024, we enrolled 29 patients with a diagnosis of primitive intra-axial focal lesions (histologically diagnosed as reported in Table 1, according to WHO 2021), who were followed at the Neurosurgical Department of the University Hospital of Florence, to perform nTMS cognitive mapping, including VSf. Patients were enrolled for the possible functional involvement of the VS network. Thirty-one surgical procedures and preoperative planning sessions were performed, since three patients were operated on twice and were tested with nTMS either for the first or second operation. One patient was excluded because of non-compliance with the neurocognitive testing, and another since he had an extra-axial lesion (histologically diagnosed as meningioma); therefore, the final number of patients was 27, and the number of nTMS procedures considered for analyses was 30 (median level of education: 12.16 ± 2 years; mean age: 50.77 ± 2 years). The gender distribution was comparable (M:F = 1:1). The histological diagnoses and anatomical locations are summarised in Table 1. Among 27 patients, 15 presented with brain lesions in the right hemisphere, 11 on the left 203 hemisphere, and 1 patient had a right parietal G3 glioma involving the splenium of the 204 corpus callosum, with contralateral extension on the left side. Twelve awake procedures were performed; two of them were aborted for anaesthesiologic issues. The final number was 10.

Seventeen were asleep procedures, where the surgical strategy was guided by the preoperative nTMS map (Table 1). The safety of cognitive nTMS was complete: no patient developed nTMS-related epilepsy. The final number of 27 patients was tested for VSf during nTMS mapping. We analysed: (I) pre- and postoperative clinical outcomes, (II) adaptation of the surgical strategy 211 on the basis of nTMS mapping and nTMS/DCS correspondence; (III) the cortical 212 nTMS and corresponding subcortical circuits.

For the surgical strategy, awake or asleep procedures were chosen on the basis of several characteristics (anaesthesiologic parameters, compliance, no preoperative deficits). At the beginning of the operation, after craniotomy and dura opening, using the neuronavigational system integrated in the microscope, nTMS points were identified and marked with a sterile tag, positioned on the cortical surface, to preserve it in asleep surgery or to compare it with DCS in awake surgery.

Globally, 55% of nTMS-positive points for VS functions were exposed in the craniotomies performed, and 75% were spared, when visible, even if at the margins of the craniotomy. We tried to avoid nTMS-positive points that were considered functionally active, but it was not possible when they were in the pathological cortical surface. Particularly, in two cases they were necessary to remove during surgery, and the postoperative functional outcome was monitored carefully.

In awake surgery, sterile tags were positioned over the nTMS-positive sites and compared with DCS spots, acquiring the spatial coordinates of these points.

Our patients were discharged 5 ± 7 days after surgery; three of them were re-operated for recurrence; one patient had a complication of surgical wound infection, which was treated with surgical toilette and antibiotic therapy. The clinical results were recorded immediately after surgery and at 30 ± 10 and 90 ± 10 days after surgery.

### 3.1. Clinical Outcome for VS Function

VSf and neglect were tested preoperatively in 27 patients, and 77.78% of procedures were performed on primary lesions. Preoperative deficits were present in 11% (3/27) of patients, whereas the majority did not have any impairment in VSf (89%, i.e., 24/27). However, in the immediate postoperative period (5 ± 2 days after surgery), we documented deficits in 40.74% of patients (11/27); 27% (3/11) of them had a preoperative impairment that worsened after surgery. These data were based on cognitive evaluations (including the bell test, OCS, and the preoperative cognitive battery). Thirty-three percent of our patients developed postoperative left neglect, since the bell test and OCS test results showed visuospatial inattention on the left side, which was the contralateral side of the brain lesion. A similar pattern was found among the bell, clock, and OCS tests, where it seemed that after 4 days, there was a deterioration in functionality that then improved over the days. As for the MMSE test, on average, it did not seem to provide satisfactory information (Figure 2A,B). We saw a temporary decrease in MMSE performance in patients with immediate postoperative USN, whose progressive recovery was documented during a 3-month follow-up period, documented as the postoperative tests results improving (90.62% in MMSE, 98.86% in the bell test, 80% in the clock test, and 98% in the OCS test) (Figure 2, Table 2). In one case, the postoperative evaluation was significantly compromised, also including linguistic–computational deficits. Nevertheless, there was a progressive improvement at 1 month after surgery and an almost complete recovery in the long-term. We calculated the percentage of functional recovery, finding that 11/27 patients (approximately 40%) experienced transient functional deficits, with a decline in performance across all VS tests, reaching a functional decrease to 80–85% for right-sided lesions and 38% for left-sided lesions. Progressive recovery was observed during the follow-up period, with clinical improvement restoring 80–98% of function, according to the previous test results.

#### Experimental VISA Results

The visuospatial integrated assessment was proposed and corrected by our neuropsychologist and speech therapist. If we compare VISA with the traditional tests for VSf, VISA is advantageous because is very easily providable to use for oncological patients, with variable levels of difficulty, depending on the type of images appearing on the screen in front of the subjects. The only caution we must adopt is the comprehension of patients, whose capability to understand and consequently perform the test must be checked in advance. Preliminary results on 20 of our patients showed that performance in this test correlated with the more classical line bisection (rho = 0.58, *p* = 0.007) and number bisection (rho = 0.60, *p* = 0.005) tasks, both also implemented on a tablet.

### 3.2. VS Network: nTMS-Based Tractography and nTMS Mapping

Tractographies were reconstructed, based on the MRI DTI sequences, using StealtViz software S8 (Stealthviz; Medtronic, Dublin, Ireland). Multiple ROIs (regions of interest) at the cortical level were identified, based on the location of specific regions, to reconstruct the three branches of the superior longitudinal fasciculus (SLFI, -II, and -III) [13,18] and then using the multiple ROIs including the nTMS points for visuospatial functioning. According to our data projected on SLFI, we recorded 168 nTMS− and 6 nTMS+ points; on SLFII, 203 nTMS− and 12 nTMS+ points; and on SLFIII, 174 nTMS− and 5 nTMS+ points. Therefore, the most represented branch of the SLF in VS circuits is SLFII (37.24% nTMS−, 52.15% nTMS+), followed by SLFIII (31.92% nTMS−, 21.74% nTMS+) and SLFI (30.82% nTMS−, 26% nTMS+), according to our preliminary data (Figure 3). Further analyses are needed to better understand the role of white matter fibre tracts in elaborating/integrating multiple functions.

### 3.3. Convergence of DCS and nTMS 

For all our patients, to validate the nTMS points (nTMS) by comparing them with direct cortical stimulation (DCS), we considered 12 awake procedures. Two were aborted for anaesthesiologic problems, so the final number was 10, with a standardised protocol of acquiring coordinates (x, y, z) for the nTMS and DCS points. We previously proposed a statistical evaluation considering the 3D brain location involved with a specific cognitive function, collecting nTMS-positive and -negative points’ spatial coordinates (x, y, z) from the neuronavigational system. The routinely used neuronavigation microscope integration (NMI) system consists of a microscope (Kinevo 900, Zeiss) combined with the StealthStation (Medtronic). During the cortical mapping in awake surgery, we highlighted DCS-positive and negative points in the neuronavigational system and collected the spatial coordinates (x, y, z). These sets of coordinates were employed in subsequent analyses, performed using specific scripts written in Python. The use of coordinates allowed us to compare the distributions of DCS and nTMS points statistically. We built up a statistical distribution in the Euclidean space, using the kernel density estimation (KDE) process, setting σ = 5 mm (the confidence level was set at α = 0.95). We finally obtained a spatial distribution of nTMS and DCS points, with the marginal probability densities fxy, fxz, and fyz. The mathematic statistical model is explained in the Supplementary Material of our previous work [6]. With these methods, a comparison between DCS- and nTMS-positive points may be more objective, reproducible, and reliable. Our analyses documented a sensitivity (Se) of 12%, a specificity (Sp) of 91.21%, a positive predictive value (PPV) of 42%, and a negative predictive value (NPV) = 66%. Accuracy was estimated at 63.7%. Although this is a very small sample of patients, on the basis of these preliminary results, we can optimise our surgical collection of data to validate our method of analysis (Table 3 and Table 4 and Figure 4, Figure 5, Figure 6 and Figure 7).

## 4. Discussion

Our work proposed the following main objectives of evaluating:(1)the clinical utility of nTMS-based preoperative planning for VSf;(2)the white matter fibre tracks underneath the VS network, analysing the predominant role of the three branches of the SLF;(3)nTMS and DCS during awake procedures to validate nTMS for cortical mapping of the VS circuits, relating cortical and subcortical data.

### 4.1. VS Functions

VS functions are fundamental to performing many activities in life. Recent efforts have been directed to investigate a wider range of cognitive and perceptual functions in neuro-oncological patients [19]. USN is a complex neurological disturbance that is not linked to sensory or motor deficits [2]. In neuro-psychology, USN impairment is usually assessed by means of line-bisection tests [20,21], line-bisection judgement tasks [7,22,23], and chimeric object and face naming tasks [24]. These tasks were first administered to post-stroke patients [25] and extended afterwards to neuro-oncological patients to control postoperative difficulties [26,27]. The line bisection judgment task [21] and the computerised line bisection task [27] are among the most commonly used methods to assess USN in awake surgery routines [28,29]. In neurosurgery, nTMS is gaining a relevant role in preoperative studies of neuro-oncological patients, with validated applications for motor and language functions [30,31], and novel ones for arithmetic calculation [6] and VSf [7]. Therefore, understanding the convergence between nTMS and DCS is pivotal for advanced cognitive mapping [4]. From a structural point of view, the application of cortical or subcortical stimulation in awake surgery led to the identification of brain areas critically involved in VSf and impairment (USN). At the cortical level, the posterior superior parietal lobe and the inferior parietal lobe constitute regions of interest for the USN [7,27,32] while, at the subcortical level, the SLFII has obtained converging evidence for its involvement in visuospatial attention [20,33].

### 4.2. rnTMS and Cognitive Tasks

To map the VS network by rnTMS, a pilot study on 10 healthy right-handed subjects was performed, using the greyscales task [3], mapping 52 cortical spots in both hemispheres. The task’s pictures showed pairs of horizontal rectangles shaded continuously from black at one end to white at the other, mirror-reversed [3,23], to test neglect. A case report [7] of a patient with a right temporal low-grade glioma applied rnTMS with line bisection and the bells test. Another study [34] on patients harbouring right parietal lesions exploited Hooper’s visual organization test (HVOT) to identify cortical areas involved in the VS network. In our experience, the same stimulation parameters (five pulses/5 Hz rnTMS) were used, and the experimental VS integrated test was useful in either the pre-, intra-, or postoperative phases to prevent USN and analyse the clinical outcome. While optimisation and validation of this test is ongoing, it appears to be a promising test to simultaneously measure and preserve visuospatial perception, visuospatial attention, non-verbal reasoning, cognitive control, and number reading.

### 4.3. Anatomic Location of VSf and the nrTMS Protocol

The VS network, thought of as capable of directing attention to spatial stimuli, relies on a fronto-temporal and parietal network [35], which may be lesioned by the tumours (48% of lesions included in our study involved more than one lobe). A bi-hemispheric mapping protocol is advisable, since reports on ipsilesional neglect are less common than those on classical contralesional neglect [3,23]. Rightward errors (or left-sided neglect) occurred significantly more often during stimulation of the right hemisphere, according to contralesional neglect, than during stimulation of the left hemisphere (ipsilesional neglect). This evidence is congruent with our preliminary results. Furthermore, the rightward errors within the left hemisphere were mainly located in the anterior parietal regions. Interestingly, this evidence was confirmed in previous studies [36]. The stimulation of the left hemisphere elicited primarily leftward errors and then contralesional neglect [3,23,37,38]. Another study [34] compared the error rate (per error type) between the right and left hemispheres: VS errors were significantly more frequent in the right hemisphere, while language-based errors were in the left hemisphere. These results are in agreement with our preliminary data.

Concerning the anatomical circuits of VS attention, a study demonstrated that the anatomical areas of brain damage resulting in neglect-like symptoms are highly dependent on the task used to diagnose them [39]. In our work, only the experimental VISA was used for rnTMS mapping, comparing the results with previous cognitive tests performed by the same patients.

### 4.4. nrTMS Mapping and DTI Tractography

DTI tractography for 20 right parietal lesions was used to compute the subcortical component of the VS network, consisting of the three branches of the superior longitudinal fasciculus (SLF) [34]. DTI reconstruction was based on a multiple regions of interest (ROI) approach, as we carried out in our study. Tractography studies demonstrated the role of the SLF in the VS network, since SLFI, -II, and -III connect the posterior parietal cortex to the dorsal and ventral frontal cortex, thus creating a complex fronto-parietal VS network. In our study, we tried to correlate the cortical and subcortical data, specifying that SLFII seems to be the most involved branch of the VS circuits, followed by SLFI and -III. Respecting the subcortical circuits is essential during surgery to avoid clinical impairment.

### 4.5. nTMS and DCS for VSf

When comparing nTMS with DCS during awake procedures, we found that nTMS still shows a high negative predictive value (66.07%). This means that negative nTMS points are more reliable for identifying “safe entry zones” which are not functionally active cortical areas, allowing us to perform corticectomy and enter the brain parenchyma, creating a safe surgical corridor to the lesion. It is reasonable to suggest that more extensive cognitive nTMS mapping data, including VSf, are needed to make meaningful comparisons with DCS, as VSf has not been thoroughly explored yet, and our study presents only preliminary data.

The parameters of nTMS stimulation were the same as those used for language and calculation mapping, with an average intensity of 100% of the resting motor threshold (rMT), with a standard deviation (SD) of 9.0% and a range of 80–120% of rMT. During awake surgeries, the stimulation intensity usually ranges from 0.5 to 3 mA to minimize the risk of seizures.

The previously described values for nTMS mapping for language and calculation [6] were higher in terms of sensitivity, specificity, and positive predictive value. We need to implement our data to confirm these results.

### 4.6. nTMS and Neglect Rehabilitation Programmes

TMS has been applied for the rehabilitation of USN after a brain lesion, mostly enrolling post-stroke patients at different stages of the disease. Following either the hypothesis of the downregulation of the intact hemisphere or of the upregulation of the affected hemisphere, different stimulation protocols were proposed, with diverging findings. As far as we know, little is known on the potentialities of TMS modulation for the recovery from USN in other populations (e.g., neuro-oncological patients). Our next goal is to investigate this point with an early rehabilitation program.


**Limitations:**
(a)The limited number of patients and surgical procedures;(b)Recurrent lesions (possible bias for clinical results);(c)VISA is an experimental test, whose value is significant for implementing the preoperative study of our patients; however, we need more patients to acquire data and compare the data with classical neuropsychological evaluations.


## 5. Conclusions

Our preliminary work shows that nTMS is valuable for explore VS functions and prevent neglect after brain tumour surgery. Our study represents a preliminary but promising experience to implement the use of nTMS in the preoperative mapping of cognitive functions.

## Figures and Tables

**Figure 1 cancers-16-04250-f001:**
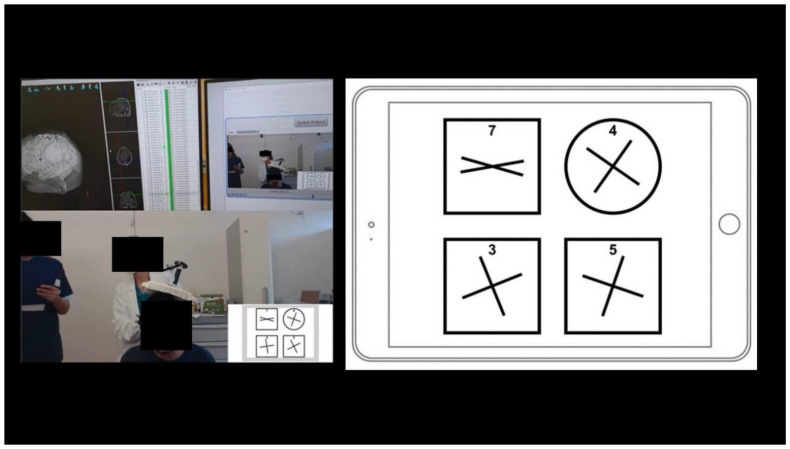
Example of a VISA test. The preoperative planning of a male patient (28 years old, harbouring a left parietal lesion) used the experimental VISA test. Participants are asked to identify the odd one out among four drawings, while disregarding the surrounding square or circular frame, and responding by spelling out the number corresponding to the selected drawing. In this example the correct answer is “7”. The test, currently under validation, aims at testing the following functions: attention, working memory, frontal inhibition, numeric recognition, geometry, and the visuospatial network. It is possible to use it (1) during the clinical evaluation phase, comparing the results with those of the classical neuropsychological test for VS attention; (2) during preoperative nTMS cognitive mapping; (3) in awake surgery during DCS; (4) and potentially for rehabilitation programmes. The advantage is that is a very easily providable test to use for oncological patients. Preliminary results on 20 of our patients showed that performance on this test correlated with more classical line bisection (rho = 0.58, *p* = 0.007) and number bisection (rho = 0.60, *p* = 0.005) tasks, both also implemented on a tablet.

**Figure 2 cancers-16-04250-f002:**
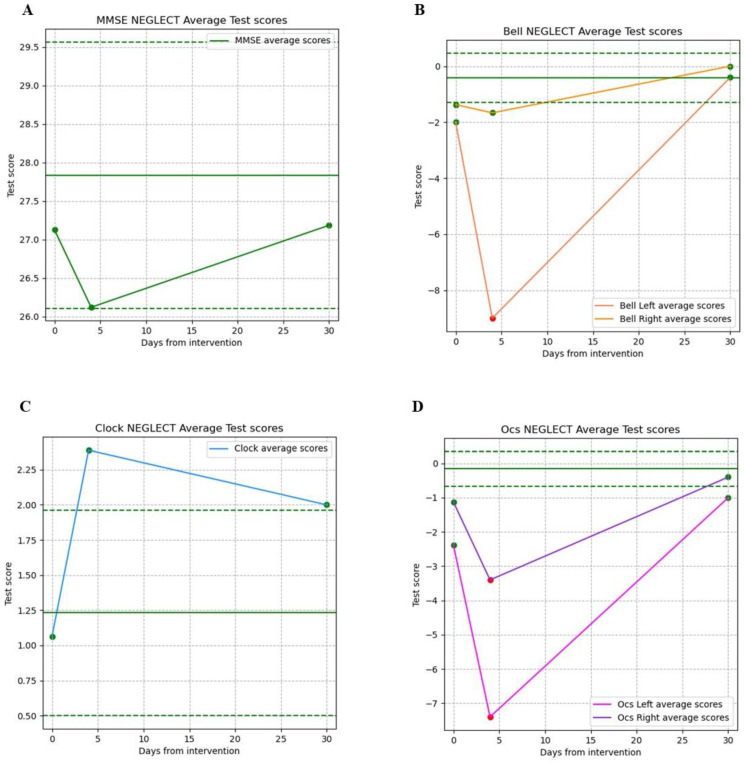
Average values of the scores for the various VSf tests. In each chart, the horizontal line represents the mean value of the test calculated on healthy patients, who do not show neglect, while the dashed lines represent the confidence interval at 1 standard deviation of the values of healthy patients. The green dots represent the mean values that are within the accepted range for the single test, while the red dots are outside the accepted ranges. (**A**) Average score values for the MMSE test for patients with pre- and postoperative neglect; (**B**) average score values for the bell test for patients with pre- and postoperative neglects, divided between the right and left hemispheres; (**C**) average score values for the clock test for patients with pre- and postoperative neglect; (**D**) average score values for the OCS test for patients with pre- and postoperation neglect, divided between the right and left hemispheres. Most patients present neglect on the left side; this is consistent with the test results in the left area in bell test (**B**) and the OCS test (**D**). A similar pattern is found for the bell (**B**), clock (**C**), and OCS (**D**) tests, where it seems that after 4 days, there is a deterioration in functionality that then improves over the days. As for the MMSE (**A**) test, on average, it does not seem to provide satisfactory information.

**Figure 3 cancers-16-04250-f003:**
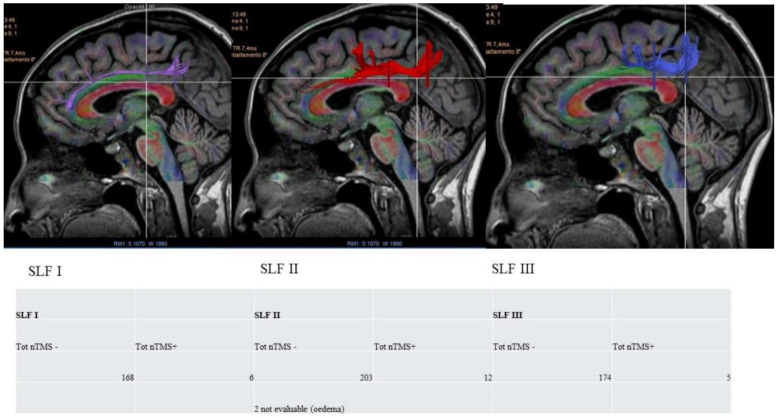
Reconstruction of SLFI, -II, and -III branches and the corresponding numbers of nTMS positive/negative points projected on each one. DTI tractography based on nTMS for VS functions allowed us to identify SLF II > SLF III > SLF I as involved in the VS fronto-parietal circuits, in line with the literature data: SLFII (37.24% nTMS−, 52.15% of nTMS+), followed by SLFIII (31.92% nTMS−, 21.74% nTMS+), and SLFI (30.82% nTMS−, 26% nTMS+).

**Figure 4 cancers-16-04250-f004:**
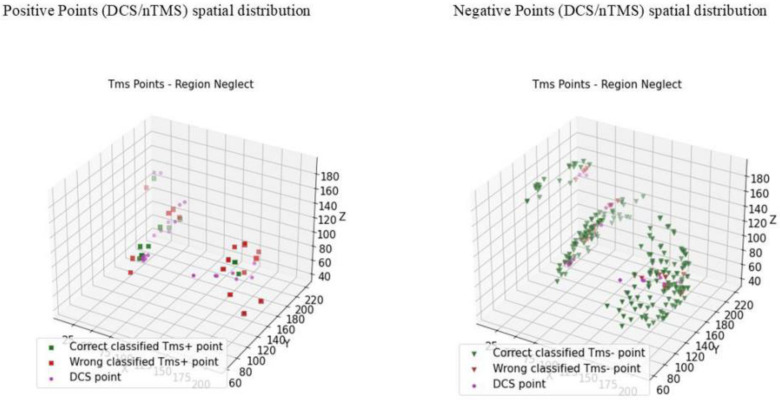
Cumulative data of patients with neglect who underwent a surgical procedure while awake and testing intraoperatively (n = 10). Left side: whole spatial distribution of nTMS- and DCS-positive points, with the marginal probability densities $f_{XY}$, $f_{XZ}$, and $f_{YZ}$ (95% confidence interval (CI); kernel density estimation process (KDE); setting σ=5 mm). Right side: whole spatial distribution of nTMS- and DCS-negative points, with the marginal probability densities $f_{XY}$, $f_{XZ}$, and $f_{YZ}$ (95% confidence interval (CI); kernel density estimation process (KDE); setting σ=5 mm).

**Figure 5 cancers-16-04250-f005:**
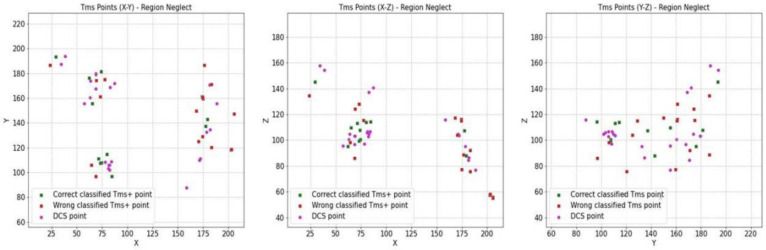
Whole spatial distribution of nTMS- and DCS-positive points, with the marginal probability densities $f_{XY}$, $f_{XZ}$, and $f_{YZ}$ (95% confidence interval (CI); kernel density estimation process (KDE); setting σ=5 mm).

**Figure 6 cancers-16-04250-f006:**
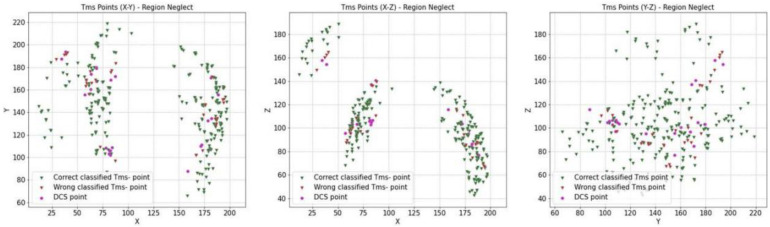
Whole spatial distribution of nTMS- and DCS-negative points, with the marginal probability densities $f_{XY}$, $f_{XZ}$, and $f_{YZ}$ (95% confidence interval (CI); kernel density estimation process (KDE); setting σ=5 mm).

**Figure 7 cancers-16-04250-f007:**
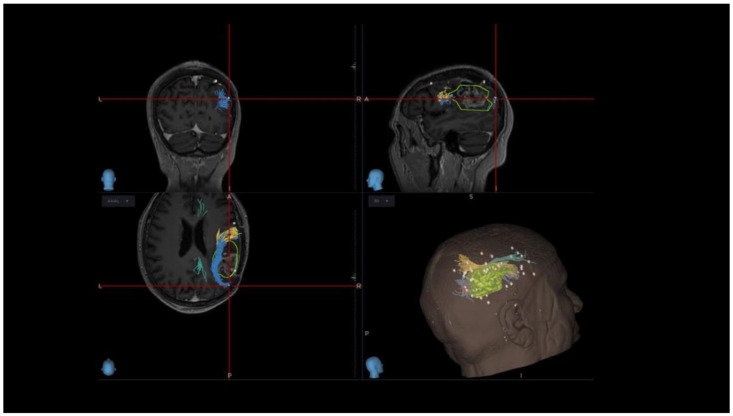
Intraoperative navigation system view, with 3D reconstruction of the brain lesion and branches of the SLF, and nTMS-positive points (violet dots) and -negative points (white dots) for visuospatial circuits, identified by the nTMS VISA test during preoperative planning. The crosshair functions of the neuronavigation system allow the acquisition of the three spatial coordinates (x,y,z) for each nTMS/DCS point, whose projections on the three branches of the superior longitudinal fasciculus were acquired and compared with those derived from the software used by our neuroradiologists, i.e., IntelliSpace Portal software (Philips).

**Table 1 cancers-16-04250-t001:** Patients’ characteristics of cognitive mapping with nTMS.

Patients	Sex	Age	Lesion	Scholarship	Awake	nTMS
*P1*	F	60	GBM, right temporo-occipital	13	+	L/C/N
*P1*	F	62	GBM, right temporo-occipital (R)	13	+	
*P2*	M	58	Metastasis (lung AdenoK), left parietal	13	+	L/N
*P3*	F	44	GBM, right parieto-occipital	15	+	L/C/N
*P4*	M	66	GBM, right temporal	8	−	L/C/N
*P5*	F	60	Cavernoma, right frontal	8	Failed	L/C/N
*P6*	F	50	Right fronto-temporo-insular	13	−	L/C/N
*P7*	M	68	GBM, left temporo-occipital	8	−	L/C/N
*P8*	M	73	GBM, right fronto-temporo-insular	8	−	L/C/N
*P9*	M	21	GBM, right fronto-parietal	13	+	L/C/N
*P10*	M	29	G2, right frontal	16	−	L/C/N
*P11*	M	29	G2, right parietal	18	−	L/C/N
*P12*	M	54	GBM, right fronto-insular	8	−	L/C/N
*P13*	F	61	G2, right frontal	13	+	L/C/N
*P14*	F	54	G2, left frontal	13	+	L/C/N
*P15*	M	52	Cavernoma, right temporal	16	+	L/C/N
*P16* (excluded)	F	50	Meningioma, left parieto-occipital	15	−	L/C/N
*P17*	M	23	Focal cortical dysplasia, left temporal	13	−	L/C/N
*P18*	F	43	G3, left parietal	16	+	L/C/N
*P19*	M	57	GBM, right temporo-insular	13	−	L/C/N
*P20*	M	33	G3, left parietal	8	+	L/C/N
*P21*	F	49	GBM, right temporo-parieto-occipital	8	−	L/C/N
*P22*	M	47	GBM, splenium corpus callosum	13	−	L/C/N
*P22*	M	48	GBM, splenium corpus callosum	13	−	
*P23*	F	43	G2, left temporo-insular	8	failed	L/C/N
*P24*	F	44	* G3, left frontal	16	+	L/C/N
*P25*	M	44	* G2, left frontal	16	+	L/C/N
*P26*	M	67	GBM, left parietal	13	+	
*P26*	M	68	GBM, left parietal	13	−	L/C/N
*P27*	F	71	* GBM, right parietal	8	−	L/C/N
*P28* (excluded)	M	73	G3, right parietal	8		L/C/N
*P29*	F	42	G3 Left parietal	13		L/C/N

**Characteristic of enrolled patients:** F = female; M = male. L = language; C = calculation; N = neglect. GBM = Glioblastoma, G2 = Glioma grade II sec WHO 2021, G3 = Glioma grade III sec WHO 2021. R = Recurrence. * Relapse of a previous glioma (no previous nTMS mapping).

**Table 2 cancers-16-04250-t002:** Average values and average percentage of functionality values obtained from tests on patients with neglect. For the bell and OCS tests, the mean values are reported, divided by the right and left hemispheres.

		Average Score	STD Score	Average Functionality (%)
*MMSE pre op*		27.13	2.72	90.42
*MMSE post op 4 D*		26.12	3.52	87.07
*MMSE post op 30 D*		27.19	1.46	90.62
*Bell pre op*	Left	−2	2.60	94.29
	Right	−1.375	1.41	96.07
*Bell post op 4 D*	Left	−9	9.97	74.29
	Right	−1.67	1.89	95.24
*Bell post op 30 D*	Left	−0.4	0.49	98.86
	Right	0	0	100
*Clock pre op*		1.06	0.17	98.75
*Clock post op 4 D*		2.39	1.54	72.22
*Clock post op 30 D*		2	0.95	80
*OCS pre op*	Left	−2.38	2.34	95.25
	Right	−1.13	1.69	97.75
*OCS post op 4 D*	Left	−7.4	11.96	85.2
	Right	−3.4	3.95	93.2
*OCS post op 30 D*	Left	−1	1.55	98
	Right	−0.4	0.8	99.2

**Table 3 cancers-16-04250-t003:** Results comparing DCS and nTMS points for neglect.

	DCS-Positive	DCS-Negative	Total
nTMS-positive	22 (TP)	38 (FP)	60
nTMS-negative	58 (FN)	476 (TN)	534
TOTAL	80	514	

Legend: TP, true positive; FP, false positive; TN, true negative; FN, false negative.

**Table 4 cancers-16-04250-t004:** Statistical results comparing nTMS and DCS points for neglect.

Sensitivity (Se)	11.93%
Specificity (Sp)	91.21%
Positive predictive value (PPV)	41.43%
Negative predictive value (NPV)	66.07%
False negative rate (FNR)	88.07%
False positive rate (FPR)	08.78%
Accuracy	63.69%

## Data Availability

Data are contained within the article.

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
