# Peer review of "Application of Navigated Transcranial Magnetic Stimulation (nTMS) to Study the Visual–Spatial Network and Prevent Neglect in Brain Tumour Surgery"

_cancers, 2024, doi:10.3390/cancers16244250_

Round 1
Reviewer 1 Report
Comments and Suggestions for Authors
This study titled -Application of Navigated Transcranial Magnetic Stimulation 2 (nTMS) to Study the Visual-Spatial Network and Prevent Ne-3 glect in Brain Tumour Surgery- is adopting a protocol to use rnTMS for preoperative planning, including VS functions for lesions potentially involving the VS network, including neurosurgical awake and asleep.
The structure is prepared well. I suggest
Introduction- at the end of this paragraph please provide goal with layout nopointed aims.
Material and methods-reference 12 cannot be just cited at the end of your procedure.
Results – all date needs statistical presentation
Please try to implement more figures, Sincerely
Reviewer 2 Report
Comments and Suggestions for Authors
The manuscript “Application of Navigated Transcranial Magnetic Stimulation (nTMS) to Study the Visual-Spatial Network and Prevent Neglect in Brain Tumour Surgery” reports an interesting study on visuo-spatial functions in neurosurgical oncological patients by using repetitive Navigated Transcranial Magnetic Stimulation. The study provides an initial yet promising demonstration of utilizing nTMS for preoperative mapping of cognitive functions. Results are exciting; however, authors are required to address followings.
1. Line 191-192; “Among 27 patients, 15 were harbouring right lesions, 11 left lesions and one patient had a bilateral G3 glioma” The statement is confusing, should be revised.
2. Line 200-201; Finally (III) cortical nTMS and corresponding subcortical circuits were analysed. It is not clear what is (III) here?
3. Line 229-230; “33% of our patients developed left neglect, i.e. presented neglect in the left hemifield, according to test results in the left area in Bell test and OCS test.” This statement is unclear. Could the author clarify what is meant by 'presented neglect in the left hemifield'?
4. Figure 2 is blurry.
5. Like 275-278; “Two were aborted for anaesthesiologic problems, so the final number was 10, with standardised protocol of acquiring coordinates (x, y, z) for n TMS and DCS points Comparison between DCS and nTMS positive points may be more objective, reproducible, and reliable. Could the authors provide further clarification on what they are trying to convey?
6. Figure 4 is so blurry, and small fonts that it’s difficult to understand.
Minor:
1. Line 121; ‘M.R.I. machine’ should be ‘MRI machine’.
2. Line 127, 183, 197, 274; ‘n TMS’ should be ‘nTMS’.
3. Line 299; neuro-on-cological should be ‘neurooncological’
Round 2
Reviewer 2 Report
Comments and Suggestions for Authors
This revised version of the manuscript responses reviewers' comments satisfactorily. The manuscript is recommended for publication.